



# Distributed faulting following normal earthquakes: reassessment and updating of scaling relations

Maria Francesca Ferrario[1], Franz Livio[1]

[1] Università dell'Insubria, Dipartimento di Scienza e Alta Tecnologia, Como, 22100, Italy

*Correspondence to*: M. Francesca Ferrario (francesca.ferrario@uninsubria.it)

**Abstract.** Coseismic surface faulting is a significant source of hazard for critical plants and distributive infrastructures; it may occur either on the primary fault, or as distributed rupture on nearby faults. Hazard assessment for distributed faulting is based on empirical relations which, in the case of normal faults, were derived almost 15 years ago on a dataset of US earthquakes. We collect additional case histories worldwide, for a total of 21 earthquakes, and we calculate the conditional

probability of distributed faulting as a function of distance from the primary fault. We found no clear dependency on the magnitude nor the time of occurrence of the earthquakes, but our data consistently show a higher probability of rupture when compared to the scaling relations currently adopted in engineering practice. We derive updated empirical regressions and show that results are strongly conditioned by the averaging of earthquakes effectively generating distributed faulting at a given distance and those which did not generate faulting; thus, we introduce a more conservative scenario, which can be

included in a logic tree approach to consider the full spectrum of potential ruptures. Our results can be applied in the framework of probabilistic assessment of fault displacement hazard.

## 1 Introduction

Surface faulting is a significant source of hazard following moderate to strong earthquakes (i.e., M > ca. 6). The quantification of fault displacement hazard is critical for the engineering design of infrastructures and land use planning

close to active faults. The avoidance criterion is usually applied for mitigating fault displacement hazard when fault strands location is certain; however, there are situations where crossing an active fault cannot be avoided (e.g., distributive infrastructures, pipelines). Moreover, faulting can occur either on the primary fault or on secondary fault strands. Currently, a probabilistic approach (i.e., PFDHA – Probabilistic Fault Displacement Hazard Analysis) is the suggested method to calculate the expected displacement due to surface faulting for siting nuclear power plants and critical facilities (e.g.,

ANSI/ANS-2.30, 2015). This approach was firstly proposed by Youngs et al. (2003) for the high-level nuclear waste repository in the Yucca Mountain, Nevada. The method is directly derived from probabilistic seismic hazard analysis (PSHA), firstly developed by Cornell (1968), which determines the annual rate of earthquakes in which a ground motion parameter exceeds a specific value, at a given location.



The earthquake approach for assessing PFDHA (Youngs et al., 2003) expresses the rate of displacement exceeding a given
value as a function of i) the annual rate of occurrence of earthquakes with a given magnitude, ii) the probability of surface
rupture along the primary fault, iii) the probability of having off-fault rupture at a given distance from the primary fault and
iv) the probability that off-fault rupture exceeds a given displacement value.

Here we focus on the conditional probability of distributed faulting, i.e., the term iii) of the general PFDHA function. While
several efforts were usually devoted in describing and measuring surface faulting along the main rupture after strong
earthquakes (e.g., Wells and Coppersmith, 1993, Pezzopane and Dawson, 1996; Field et al., 2015), less information is
available on distributed faulting. Following the pivotal work of Youngs et al. (2003) on normal faults, the PFDHA approach
was applied to strike-slip (Petersen et al., 2011) and reverse faults (Moss and Ross, 2011; Boncio et al., 2018; Nurminen et
al., 2020). Regional datasets were built as well (e.g., Takao et al., 2013; Inoue et al., 2019 for Japan strike-slip and reverse
faults). Slightly different methods and procedures were adopted for developing empirical relations of the probability of
faulting. The decrease of the probability of faulting with distance from the primary fault was unequivocally identified by all
the studies; for other driving parameters, a general consensus has still not be achieved: for instance, a magnitude dependency
was included by Youngs et al. (2003), but not by Petersen et al. (2011). The hanging wall and footwall are considered
separately by most of the authors dealing with dip-slip ruptures (but not by Takao et al., 2013).

The role of the local structural setting, fault architecture at depth and near-surface geology (e.g., cut lithologies, overburden
load) has been highlighted after the analysis of recent earthquakes, which caused a complex pattern of rupture at surface
(e.g., Bray et al., 1994; Milliner et al., 2015; Teran et al., 2015).

Concerning normal faults, the reference paper for PDFHA is still the work by Youngs et al. (2003), which was published
almost 20 years ago and analyzed a dataset of US earthquakes. Since then, additional datasets were acquired (see Baize et al.,
2019 for a comprehensive review on the efforts in building a unified database of fault displacement), and a general
underestimation of the Youngs' relations in the far-field was recently pointed out (Ferrario and Livio, 2018). In this work,
we present data and improved regression equations for the conditional probability of off-fault rupture. We supplemented the
earthquakes already analyzed by Youngs et al. (2003) with additional case histories, for a total of 21 earthquakes. This figure
more than doubled the dataset considered in the original work.

We provide an updated version of the conditional probability of distributed faulting as a function of distance from the
primary fault. Our results broadly agree with those of Youngs et al. (2003), but systematically show a probability of
occurrence higher than expected. We thus introduced a conservative scenario, to fully account for the range of observed
probabilities, that can be handled in a logic tree approach, as commonly done for PSHA studies.



## 2 Materials and methods

### 2.1 Dataset

We analyzed 21 events occurred between 1887 and 2016, ranging in magnitude (Mw) from 6.0 to 7.5. Nine of them are from the Western US and were already analyzed in terms of FDHA by Youngs et al. (2003), which in turn used data compiled by Pezzopane and Dawson (1996). Data for the remaining 12 earthquakes were collected from scientific literature or available databases (i.e., SURE Project, Baize et al., 2019). Table 1 lists the main details and data source for each earthquake.

The considered events occurred in different countries and all the 5 continents are represented with at least one case history: 9 earthquakes are from the US, 6 from Italy, 2 from Greece and 1 from Kenya, Turkey, New Zealand and China (Fig. 1). The geographical distribution of the events reflects either the regional seismotectonic setting, or the availability and accessibility of data: the US, Italy and Greece are frequently hit by normal-faulting events; other regions, like China, are less represented in our database despite a strong earthquake activity. For the first time, an event from the African continent is included. We

stress that the addition of events from different tectonic provinces and climatic conditions will enable a better characterization of distributed faulting, resulting in an overall more reliable scaling relation. In this sense, we follow the recommendation by Baize et al. (2019) and uploaded a shapefile with the rupture sections for the 14 events, not already available in the SURE database, in an online repository (see Data Availability Sect.).

The earthquakes' magnitude and year are plotted in the inset of Fig. 1: the mean magnitude of the events analyzed by

Youngs et al. (2003) was 6.86, whereas the mean magnitude of the additional events is 6.58. The additional case histories are generally more recent in time with respect to the ones considered by Youngs, with the exception of the 1915 Fucino (Italy) and 1928 Laikipia-Subukia (Kenya) events. A particular effort was devoted to the collection of data on M < 6.5 earthquakes; they were not adequately represented in the Youngs' database, but they pose a significant threat to society, being much more frequent than stronger earthquakes (Baize et al., 2019).

### 2.2 Methods

Our methodological workflow is illustrated in Fig. 2a. The input data are shapefiles mapping the traces of surface ruptures; these can be already available in a ready-to-use form (7 events included in Baize et al., 2019), or were created from georeferencing and digitization of maps published in the scientific literature (14 events). We handled the maps with the best possible accuracy; as a general rule, the digitization of surface ruptures was performed at a scale ranging from 1:50.000 to

1:10.000, depending on the accuracy of the original maps.

We then divided the ruptures in primary and distributed faults; "primary" faulting shows longer continuity and higher displacement with respect to distributed faults and corresponds to the surface expression of the rupture along the seismogenic source at depth. Consequently, we classified as "distributed faults" (DF) all the ground breaks along structures either connected or not to the main fault, which occurred in response to principal faulting (e.g., Youngs et al., 2003;

ANSI/ANS-2.30, 2015). We discarded only ruptures explicitly referred to gravitational phenomena, according to the reports.



Different approaches have been adopted in the literature for defining the ruptures to be processed in further analyses: for instance, Petersen et al. (2011) discarded the "triggered" ruptures (i.e., not structurally connected to the primary fault), while Nurminen et al. (2020) ranked ruptures due to reverse earthquakes into different categories (ranked from 1 to 3). We did not attempt to further categorize the distributed ruptures in our database due to the lack of reliable information for some of the
case histories; moreover, from an engineering perspective the occurrence of DF itself is much more relevant than the nature of the triggering process (e.g., Youngs et al., 2003).

The conditional probability of distributed faulting was computed following the earthquake approach, as defined by Youngs et al. (2003). This was the first work to introduce PFDHA and can be categorized as a gridding approach (Nurminen et al., 2020). We derived a raster of the Euclidean distance from the primary fault with a 500-m grid resolution and counted the
number of pixels for each distance class. The conditional probability of faulting (P) is defined as the number of pixels containing distributed faulting divided by the total number of pixels in each distance class (Fig. 2a) and was calculated separately for the hanging wall and footwall blocks.

Firstly, for each event, we computed the probability of DF as a function of distance from the primary rupture - P(x). We also investigated the possible influence of magnitude range and year of occurrence (i.e., historical or modern events), assessing
whether to include or not these parameters in the further analyses (see Sect. 2.3). Then, we calculated the mean value for each distance class and fitted empirical regression to the dataset. We tested different functional forms, following those ones proposed in the literature (i.e., power form – Petersen et al. (2011); exponential form – Youngs et al. (2003)); in the following we provide the fitting coefficients for Eq. 1, a functional form equivalent to the one used by Youngs et al. (2003), because it consistently performed better in terms of fitting.

$$P(x) = \frac{e^{(a+b*(\log(x+c))}}{1+e^{(a+b*(\log(x+c))}}$$  (1)

Where x is distance from the primary fault, *a*, *b* and *c* are fitting coefficients.

## 2.3 Assumptions and limitations

Overall, PFDHA analysis requires a significant amount of subjective choices and a common methodology, or a thorough comparison of different methods, is still lacking. The gridding method adopted in the current research is not devoid of limitations, which we address in the following. Nevertheless, our primary goal was to assess the performance of the only scaling relation available for normal faults, and to update such regression with new case histories. For this reason, we basically replicated the work by Youngs et al. (2003).

A first issue regards the scale, resolution and completeness of the original maps and data. For all the selected events, detailed rupture maps are available; nevertheless, a large variability in the quality of data is present, since we investigated events occurred between 1887 and 2016. An increasing quality of the reported data for more recent events could be expected: modern technologies and integrated approaches encompassing extensive fieldwork and remote sensing (e.g., InSAR, optical



correlation techniques etc.) can capture ground deformation of few centimeters, which in the past could have easily gone
undetected (see Livio et al., 2017 for a more detailed discussion). A working hypothesis is that older events are characterized
by a higher epistemic uncertainty due to less reliable technologies or natural censoring of smaller displacements if
measurements are taken long after the earthquake occurrence (e.g., Stirling et al., 2002). The variability of modern events
should indeed be aleatory. We explored this issue by dividing the dataset in two subsets, namely events occurred in the XXI
century (4 events) and those occurred before that date (17 events).

A second issue is related to the definition of the primary fault, which in turn affects the computation of distances. The
delineation of the primary fault can be straightforward in some rupture sections but can be more complex in other sections
(Petersen et al., 2011), where the surface rupture is discontinuous or structural complexities (multiple parallel strands, gaps
between ruptures) are present. Figure 2 shows examples from the Gulf of Corinth (Greece) and Edgecumbe (New Zealand)
events: in the first case, the primary fault shows a relatively simple trace at the two ends, while in the central part a complex
pattern of ruptures is present. The Edgecumbe event ruptured the Edgecumbe fault and 10 secondary segments. In the
assessment of the primary fault, we referred to the fault plane modeled by Beanland et al. (1990), which defines the
Edgecumbe fault as the main fault rupture; the Onepu fault (see Fig. 2c) lies along-strike of the Edgecumbe fault, but shows
a much smaller displacement (maximum values of 2.5 m and 26 cm, respectively) and thus is considered as a secondary
rupture.

The issues defined above are not trivial, because the gridding method implicitly assumes that the mapping data for the
ruptures are complete, and results depend on the grid size. The grid size of 500 m is quite coarse, to compensate for the
possible underestimation due to incomplete mapping (Youngs et al. 2003). Other approaches have been explored in the
literature, such as to consider different grid sizes (e.g., Petersen et al., 2011). More recently, a "slicing" approach has been
introduced in the analysis of reverse earthquakes (Boncio et al., 2018; Nurminen et al., 2020); this method makes no
assumptions on the completeness of the database and does not depend on the grid size.

Finally, in case of multiple earthquakes in few days/months (e.g., 1981 Gulf of Corinth: Jackson et al. 1982; 2016 Central
Italy: Brozzetti et al., 2019), it may not be possible to attribute each surface rupture to its causative event. Repeated rupture
of the same fault strand is certainly a theme to be investigated, but we believe that this kind of uncertainty does not heavily
affect our results.

**3 Results**

Firstly, we computed the conditional probability of distributed faulting for each single event; then, we explored the role of
magnitude and year of occurrence as factors affecting this value.



### 3.1 The role of magnitude

Firstly, we grouped the case histories into magnitude classes (i.e., M < 6.5; 6.5 ≤ M < 7.0; M ≥ 7.0). Figure 3 shows the
conditional probability of faulting as a function of the distance from the primary fault. Positive values correspond to the
hanging wall, whereas negative values refer to the footwall. Each symbol represents P(x) at a specific distance for a single
earthquake. Points on the x-axis indicate that no distributed faulting occurred in the given distance class; black asterisks
show the mean values. The probability of rupture drops off quickly while moving farther from the main fault, with a steeper
decrease in the footwall than in the hanging wall. The ratio between the probability of faulting in the 0-500 m and 1000-1500
m classes is 7:1 for the hanging wall and 14:1 for the footwall, pointing to a fundamental difference between the near and far
fields. Some peaks in the far-field are visible as well (e.g., in Fig. 3a at distance of 7-8 km in the hanging wall).

Figure 3b shows the stacked values of conditional probability, with earthquakes ordered according to magnitude; again, it is
clear the decrease of P(x) with distance, but no clear trend is apparent for increasing magnitude values. Thus, in the
following analyses we do not explicitly include a magnitude term in the scaling relations; moreover, it is important to
highlight the relatively small sample set and that the magnitude determination for the older events (which go back to the end
of the XIX century) may bear a significant degree of uncertainty.

The mean values shown in Fig. 3 correspond quite well with the scaling relations by Youngs et al. (2003), but we underline
that mean values derive from balancing a dichotomous variable, because distributed rupture can either occur or not occur. On
one hand, earthquakes actually producing distributed faulting show probabilities much higher than the mean value; on the
other hand, several earthquakes do not produce faulting at all at a certain distance from the primary fault. Figure 4 better
clarifies this point: we calculated the percentage of earthquakes generating distributed faulting for each distance class
("events with no DF" corresponds to the points on the x-axis in Fig. 3). Most earthquakes produce distributed faulting in the
first kilometers from the primary fault, while only 30% generate surface faulting at 7 km in the hanging wall, and this value
decreases to 3 km in the footwall. Another aspect worth mentioning is that real data are constrained down to probabilities as
low as ca. 10-2 while the extrapolation to lower probabilities is purely theoretical (see "No data" field in Fig. 3a).

### 3.2 The role of the dataset age

We then grouped the case histories according to the year of occurrence, i.e., 17 earthquakes occurred before year 2000 (mean
magnitude: 6.75) and 4 more recent earthquakes (mean magnitude: 6.47). We tested the hypothesis that older earthquakes
may show lower probability of distributed faulting due to the incompleteness of dataset or less reliable measures. However,
we found no systematic bias between the two subsets and any clear pattern arises when stacking the events according to the
year of occurrence (Fig. 5).



### 3.3 Deriving scaling relations: regular and conservative scenarios

In the following we derive the equations of the conditional probability of distributed faulting as a function of distance from the primary fault. We calculate the mean probability values for each class distance and fit the data with Eq. 1. When trying to

fit all the data, we found that the functions do not extrapolate well in the vicinity of the primary fault. This drawback was already pointed out by Petersen et al. (2011), and thus we followed their suggestion of excluding the first two off-fault rupture probability measurements, which in our case correspond to distances up to 1 km from the primary fault. The obtained regression coefficients and good of fit parameters are provided in Table 2, while the curves are plotted in Fig. 6. The overall pattern matches the results of Youngs et al. (2003), but we note that observed data are consistently higher than predicted by

their equations, in particular in the hanging wall.

As pointed out earlier, computing the mean value of all the earthquakes for each distance bin results in the average of two end members: events without distributed faulting at a given distance; and events that generated distributed faulting at that distance. The conditional probability of faulting is zero for the former category, while it is variable, within a certain range, for the latter and it comes out that, when occurring, the probability is much higher than the mean value. For this reason, we

introduced a conservative scenario, computing the conditional probability only for those data that recorded distributed faulting at a given distance. The points are shown in Fig. 6, together with the curves obtained in the same way as for the regular scenario. The relevance of this conservative scenario lies in the fact that for some projects it may be necessary to consider the worst-case scenario or return period longer than the norm (Wells and Kulkarni, 2014; Cline et al., 2018).

Between 1 and 5 km from the primary fault, the conservative scenario predicts a probability of faulting that is ca. 3 times the

regular scenario, while at about 10 km from the primary fault the conservative scenario is one order of magnitude higher than the regular one. The residual plots (observed minus predicted values, Fig. 6b) show that for the regular scenario observed values better match with the predicted values, with a higher discrepancy at 1-3 km in the hanging wall. The conservative scenario shows a higher dispersion of the values, resulting in overall higher residuals and lower goodness of fit parameters (Table 2). We remark that the range of applicability of the equations is 1 to 15 km in the hanging wall and 1 to 10

km in the footwall; extrapolation beyond such limits should be gingerly considered.

### 4 Discussion

We compute the conditional probability of distributed faulting for dataset of 21 normal faulting earthquakes and provide updated empirical relations assessing the decrease of faulting with distance. Our data show a very steep decrease in the probability of faulting at distances higher than 1 km from the primary fault, pointing to a fundamental difference between

near and far-fields. The pattern of fault rupture in the near field has been recently explored by several authors, by means of observation of actual faulting, numerical or analytical models (e.g., Fletcher and Spelz, 2009; Teran et al., 2015; Gold et al., 2015; Loukidis et al., 2009; Treiman, 2010). Several factors have been pointed out as conditioning the surface expression of faulting, including type of fault movement, fault dip, amount of displacement, geometrical complexity of fault traces, rock





type, thickness and nature of the materials above bedrock (Bray et al., 1994; Avar and Hudyma 2019). The integration of

modern field and remote technologies allows to capture the finer details of surface rupture; a geologically sound interpretation of such data is pivotal to the understanding of surface rupture, which in turn is essential for engineering design and eventually mitigation measures. The regressions presented in the current work derive from a global dataset; we stress that, if site-specific data are available, they should be properly considered. For instance, a different behavior between Japan and US ruptures have been identified (Inoue et al., 2018; Petersen and Chen, 2018; Suzuki and Annaka, 2018); if such

variations are systematic, they can be to some extent predictable and thus help in assessing the hazard.

The updated empirical regressions broadly confirm the results obtained by Youngs et al. (2003), highlighting the soundness of the proposed approach and its wide applicability in different regions worldwide. The addition of new case histories allows just an incremental improvement in the equations. The conditional probability of faulting does not show dependency on magnitude or time of occurrence, thus allowing to combine all the events in a single dataset for obtaining empirical

regressions. The lack of dependency on magnitude can be explained by the fact that we analyze a global dataset, whereas Youngs et al. (2003) focused their work on earthquakes occurred in the Western US. Moreover, it has been shown that the choice of the grid size has a much larger effect than the moment magnitude of the events (Suzuki and Annaka, 2018; Takao et al., 2018).

Concerning the year of occurrence, in a previous analysis of Italian Apennines events we claim that older events and maps

may be affected by higher epistemic uncertainties (Ferrario and Livio, 2018). In the current study, we tested again this hypothesis and found no systematic difference between earthquakes occurred several decades apart; thus, we argue that the issue of data completeness is not simply a matter of the time of occurrence, but is influenced by other factors like the territorial and climatic setting (e.g., arid vs humid climate) and the potential of preservation of surface ruptures.

The most striking observation when analyzing the updated regressions is the systematic higher probabilities of faulting, in

particular in the hanging wall, with respect to the original formulation by Youngs et al., (2003). The output of the analysis is strongly influenced by the bins where no distributed faulting was observed (i.e., points on the x-axis in Fig. 3 and 5). In order to account for the full spectrum of potential occurrences, we introduce a conservative scenario, where conditional probabilities of faulting are computed without the bins with no distributed faulting. This scenario results in higher expected probabilities, and the residual plot (Fig. 6b) shows that observed values are lower than expected between 3 and 7 km in the

hanging wall, whereas are higher than expected at distances of 7-12 km. A possible explanation to this behavior may be the structural architecture of normal faults, which commonly show a horst and graben setting: the sector 3-7 km may correspond to the block which undergo lowering and/or tilting, and the sector 7-12 km should correspond to the location of the antithetic structure. This hypothesis can be tested on well-documented case histories, where detailed information is available on the structural architecture of the shallow subsoil. The position at surface of the antithetic structure is driven e.g., by the change in

dip of the primary fault at depth (e.g., Caskey et al., 1996); this points to the possibility of introducing deterministic constraints in the estimation of the expected distributed faulting. Consistently, we here point out that the use of elastic dislocation models of deformation (e.g., Okada, 1985) and, in turn, of induced Coulomb stress transfer on receiving pre-





existing faults (e.g., Toda et al., 2011) may be useful for better predicting the probability of DF (e.g., Gurpinar et al., 2017, Livio et al., 2017), especially where the current models show higher residuals, e.g., at ca. 7-12 km in the hanging wall.

We claim that the regular and conservative scenarios can be seen as alternative branches in a logic tree; the relative weights of the two branches can be tuned according to the degree of conservativity of the project and to professional judgement. We argue that the conservative scenario can be particularly important in the analysis of normal faults, for two main reasons: i) normal faults have a higher probability of generating surface faulting along the primary fault when compared to strike-slip and reverse faults (Moss and Ross, 2011; Suzuki and Annaka, 2018; Takao et al., 2018); ii) the conditional probability of

distributed faulting for normal events is much higher than for strike-slip and reverse earthquakes. The latter point is better illustrated in Fig. 7, where we compare our regressions for normal faulting with the original formulation of Youngs et al. (2003) and the one proposed by Takao et al. (2018) for strike-slip and reverse Japanese events, developed using the same grid size of 500 m.

## 5 Conclusions

We develop an updated regression for the conditional probability of distributed faulting as a function of distance. We propose two alternative scenarios, to take into account the wide range of ruptures: the regular scenario computes the average value of all the analyzed events, while the second scenario provides a more conservative estimate. We believe that a periodic update of the database allows to derive more robust relations: for this reason, measures of distributed faulting and data implementation in a common framework (e.g., Baize et al., 2019) should be a standard practice following every ground-

rupturing event. In the effort of common data mining, the shapefiles of ruptures for the new case histories and the table with conditional probability of faulting derived in the present research are made available in an open repository (see Data Availability Sect.).

Here we focus on the conditional probability of faulting and do not consider the amount of faulting (i.e., we treat faulting as a binomial variable YES/NO). Future research should consider also the amount of displacement, to fully implement our

result in a PFHDA perspective. The assessment of uncertainties in PFDHA analyses is not yet fully developed and methodological choices have a substantial effect on the total hazard (Moss and Ross, 2010; Treiman, 2010; Wells and Kulkarni, 2014; Cline et al., 2018); logic trees can be used to consider the full distribution of rupture characteristics. Finally, a critical comparison of different methods and procedures should be pursued and the comparability with the original works (e.g., Youngs et al., 2003) should be guaranteed.

**Data availability**

The shapefiles with rupture sections and trace of the primary fault, and a table with the conditional probability of faulting as a function of distance are accessible at https://zenodo.org/record/4323267#.X9jzlLN7lPZ DOI: 10.5281/zenodo.4323267



Author contribution: MFF designed the workflow, analyzed the data and drafted the manuscript. FL analyzed the data and reviewed the manuscript.

Competing interests: The authors declare that they have no conflict of interest.

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

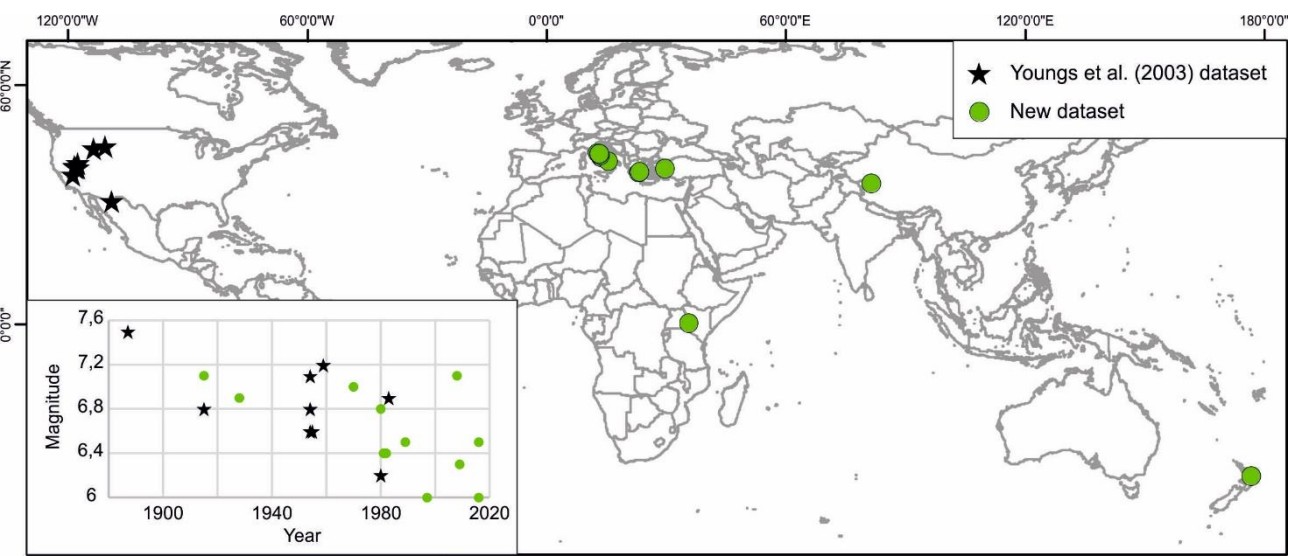

**Figure 1: Location of the analyzed events. The plot in the inset shows the distribution of the events according to magnitude and time of occurrence**





**Figure 2: a) Methodology for computing the conditional probability of faulting using the gridding approach. FW: footwall, HW: hanging wall. b) example of complex trace at surface following the 1981 Gulf of Corinth (Greece) earthquake; ruptures are digitized from a map published by Jackson et al. (1982). C) example of complex trace at surface following the 1987 Edgecumbe (New Zealand) earthquakes; ruptures are from the SURE database (Baize et al., 2019).**




**Figure 3: a) Conditional probability of faulting as a function of distance from the primary fault. Earthquakes are grouped according to their magnitude (M < 6.5; 6.5 ≤ M < 7.0; M ≥ 7.0); colored symbols represent the values for each earthquake and distance class, black asterisks are the mean values. Black lines are the scaling relations by Youngs et al. (2003; their Eq. 7). The "No data" field illustrates the region where no empirical observations are present, and probabilities are extrapolated by fitting. b)**
**comparison between the 21 earthquakes, ranked according to magnitude; each line represents a single event.**



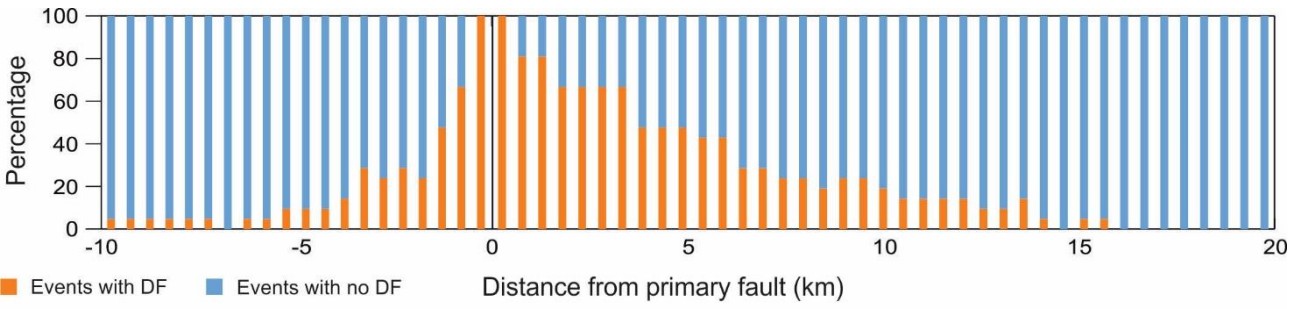

**Figure 4: percentage of earthquakes showing distributed faulting vs no faulting for each distance class.**

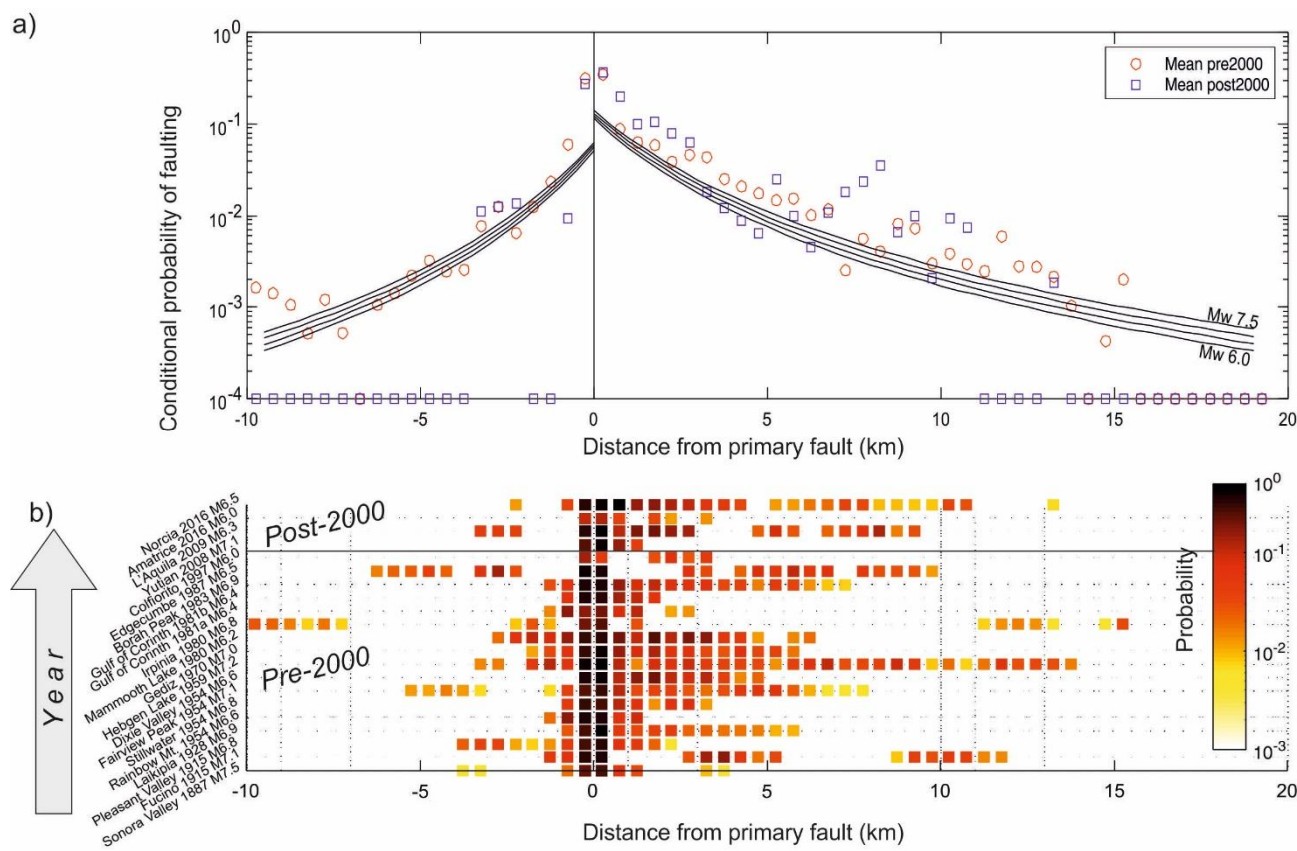

**Figure 5: a) Conditional probability of faulting for the subset of pre-2000 and post-2000 events (n. 17 and 4, respectively); colored symbols represent the mean values. Black lines are the scaling relations by Youngs et al. (2003; their Eq. 7). The "No data" field illustrates the region where no empirical observations are present, and probabilities are extrapolated by fitting. b) comparison between the 21 earthquakes, ranked according to the year of occurrence; each line represents a single event.**




**Figure 6: a) Conditional probability of faulting as a function of distance from the primary fault. Black lines are the scaling relations by Youngs et al. (2003; their Eq. 7). Colored lines are the equations proposed in the present research ("regular" and**





"conservative" scenarios). B) Proposed equations and 95% confidence bounds (y-axis: linear scale). C) Residual plot (observed – predicted).

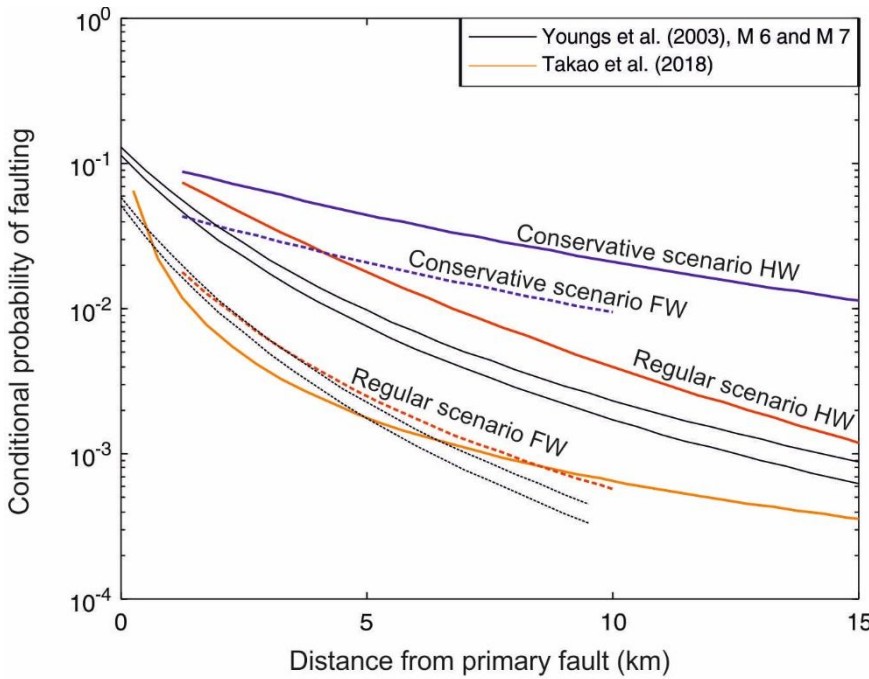

**Figure 7: Comparison between the conditional probability of faulting for normal events obtained in this study and the ones by**
**Youngs et al. (2003; the curves for M 6 and 7 are shown as a reference) and for strike-slip and reverse events in Japan (Takao et al., 2018); full lines: hanging wall, dashed lines: footwall.**


**Table 1: Details on the 22 earthquakes analyzed in the present study. Date expressed as year, month, day. Reference list the source of the map of surface ruptures; * in the reference column highlights the events already included in the work by Youngs et al. (2003).**

| ID | Date (yyyy-mm-dd) | Location | Country | M | Reference |
|----|----|----|----|----|----|
| 01 | 1887-05-03 | Sonora Valley | US | 7.5 | Baize et al. (2019) * |
| 02 | 1915-01-13 | Fucino | Italy | 7.1 | Amoroso et al. (2016) |
| 03 | 1915-10-03 | Pleasant Valley | US | 6.8 | Baize et al. (2019) * |
| 04 | 1928-01-06 | Laikipia-Subukia | Kenya | 6.9 | Ambrasyes (1991) |
| 05 | 1954-07-06 | Rainbow Mountain | US | 6.6 | Pezzopane and Dawson (1996) * |
| 06 | 1954-08-23 | Stillwater | US | 6.8 | Pezzopane and Dawson (1996) * |
| 07 | 1954-12-16 | Fairview Peak | US | 7.1 | Baize et al. (2019) * |
| 08 | 1954-12-16 | Dixie Valley | US | 6.6 | Baize et al. (2019) * |





| 09 | 1959-08-18 | Hebgen Lake | US | 7.2 | Baize et al. (2019) * |
|----|-----------|-------------|-----|-----|----------------------|
| 10 | 1970-03-28 | Gediz | Turkey | 7.0 | Ambraseys and Tchalenko (1972) |
| 11 | 1980-05-25 | Mammoth Lake | US | 6.2 | Baize et al. (2019) * |
| 12 | 1980-11-23 | Irpinia | Italy | 6.8 | Ferrario and Livio (2018) |
| 13 | 1981-02-25 | Gulf of Corinth | Greece | 6.4 | Jackson et al. (1982) |
| 14 | 1981-03-04 | Gulf of Corinth | Greece | 6.4 | Jackson et al. (1982) |
| 15 | 1983-10-28 | Borah Peak | US | 6.9 | Baize et al. (2019) * |
| 16 | 1987-03-02 | Edgecumbe | New Zealand | 6.5 | Beanland et al. (1989) |
| 17 | 1997-09-26 | Colfiorito | Italy | 6.0 | Ferrario and Livio (2018) |
| 18 | 2008-03-20 | Yutian | China | 7.1 | Xu et al. (2013) |
| 19 | 2009-04-06 | L'Aquila | Italy | 6.3 | Ferrario and Livio (2018) |
| 20 | 2016-08-24 | Amatrice | Italy | 6.0 | Ferrario and Livio (2018) |
| 21 | 2016-10-30 | Norcia | Italy | 6.5 | Ferrario and Livio (2018) |

**Table 2: Data fitting and goodness of fit parameters for the traditional and conservative scenarios; regressed parameters, with 95% confidence bounds, are indicated. HW: hanging wall, FW: footwall, SSE: sum of squares due to error, RMSE: root mean squared error.**

| Model | a | b | c | R2 | Adj. R2 | SSE | RMSE |
|-------|-----|-----|-----|-----|---------|-----|------|
| HW, regular scenario | 11.87 (-9.622, 33.37) | -5.687 (-11.49, 0.1166) | 11.3 (-3.849, 26.46) | 0.9742 | 0.9727 | 0.0003149 | 0.003043 |
| FW, regular scenario | 0.01478 (-10.76, 10.79) | -2.95 (-6.961, 1.061) | 2.644 (-4.51, 9.797) | 0.8905 | 0.8817 | 6.118e-05 | 0.001564 |
| HW, conservative scenario | 6.385 (-17.53, 30.3) | -3.237 (-9.267, 2.793) | 13.53 (-22.16, 49.22) | 0.8335 | 0.8237 | 0.003727 | 0.01047 |
| FW, conservative scenario | 7.938 (-67.11, 82.99) | -3.845 (-22.26, 14.57) | 16.43 (-86.5, 119.4) | 0.6509 | 0.623 | 0.001788 | 0.008457 |