# Peer review of "Conditional probability of distributed surface rupturing during normal faulting earthquakes"

_Solid Earth, 2020_

## Referee Comment (RC1) · Anonymous Referee #1 · 1 Feb 2021

This manuscript provides updated the conditional probability of the distributed rupture of the PFDHA model with compiled the distributed ruptures of the global normal faulting data. The PFDHA model of Youngs et al. (2003), which was based on the normal fault in the US has been used for the normal faulting PFDHA. The developments of normal faulting PFDHA model are required by new data set. The descriptions of the principal and distributed faults, and detailed distributed fault data, which are provided in this manuscript, are good references for the future development of the normal faulting PFDHA model, although the distributed fault displacement attenuation equation and PFDHA evaluations based on new compiled data are not available. This manuscript focused on the probability condition of distributed rupture. The processes of constructing the conditional probability, difference from previous models and possible reasons for differences. I would like the authors to several explanations.

Please see comments for detail. I hope these comments will be helpful.

Here are the comments that I would like the authors to explain for readers with interest of PFDHA.

1. The conditional probability of the distributed rupture

1-1. Why did the authors update only the conditional probability, not distributed fault displacement attenuation relation? 1-2. Petersen et al. (2011), who uses the power function, used linear interpolation as mentioned (l. 185-186). On the other hand, Youngs et al. (2003) and Takao et al. (2013), who use the exponential function used same to this paper, do not exclude the near range from the principal fault. Why do the authors need to exclude the range of 0-1km from the principal fault?

1-3. If your conditional probability excludes the distance range of 0-1km from the principal fault, I would like the authors to describe the calculation of the probability in this vicinity as Petersen et al. (2011).

2. As the authors mentioned (l. 217-220), conditional probability is obtained from the global data set. Is this the reason for the greater probability than that of Youngs et al. (2003)? In other words, is Youngs et al. (2003) used for the US PFDHA and is the conditional probability of this study used for the non-US PFDHA?

Here are the minor comments.

3. Title

It is difficult to understand the detail contents from the title. 'normal earthquake' is expressions that is rarely seen for me. Does 'scaling relation' mean a conditional probability?

4. Typo?

FDHA -> PFDHA? (l. 63)

5. Caption of Table 1

22 earthquakes may be 21 earthquakes.

5. Eq.1 (l. 112)

Please add unit of x.

6. Figure 6(a)

Why is the yellow-colored range near the main fault in the figure different between the hanging wall and foot wall sides?

---

## Referee Comment (RC2) · Anonymous Referee #2 · 16 Feb 2021

Maria Francesca Ferrario and Franz Livio

**Anonymous Referee #2**

Review of the paper: Distributed faulting following normal earthquakes: reassessment
and updating of scaling relations by M. F. Ferrario and F. Livio

The paper presents new (updated) empirical regressions for distributed surface ruptur-
ing during normal faulting earthquakes based on an updated dataset of normal faulting
case studies. The results have implications for probabilistic fault displacement hazard
analysis (PFDHA).

The basic work on PFDHA is Youngs et al. (2003), where the empirical regressions for
distributed faulting were based on a limited number of normal faulting historical surface
ruptures. Since then, there are not published works that implement such empirical re-

gressions for normal faulting. The work presented here is certainly of interest in the international community working on PFDHA, and in general in the Solid Earth community interested in natural hazards. Therefore, in my opinion the paper deserves to be published.

The paper is well organized and well written. The figures are good and self-explanatory. I appreciate that the basic data are made available as shapefiles (some comments below).

Please, consider the following comments during the revision of the paper:

1) Nomenclature: Primary vs Principal. The reference literature on PFDHA uses this nomenclature: Principal and Distributed (Youngs et al 2003; Petersen et al., 2011). Though this is a very minor comment, it would be nice if all the specialized literature will use the same nomenclature.

2) Method (gridding) (Lines 100-102, Fig 2a). In order to make the results reproducible, can you be more detailed in describing the geometry of the grid and the method for calculating distances? Did you consider the same maximum distance from PF for all the events? How far from PF (20 km?)? Which criterion guided the choice? Was the grid (and the sides of the squares) always horizontal/vertical? Or rotate with fault strike? PF-distance: is that the shortest distance between the PF line and the centre of the cell? Adding a real case in Fig. 2a can be explicative and help the reader.

3) Extrapolation of regressions (lines 174-175). Why did you extrapolate away from observations?

4) Role of dataset age and M. You found no systematic bias between the pre- and post-2000 datasets (lines 179-181). Looking at your data, I agree with this observation. But I suspect that you do not see the differences in the analysis because of the small number of modern data compared to older data and the coarse grid size, which smooths differences. I suspect that this can influence the possible dependency on M, as well. I

think this point (possible bias due to methodology of analysis) should be addressed in the discussion.

4) Distributed faulting in the near-field (< 1 km). The regressions are cut at distances shorter than 1 km for mathematical reasons that I can understand. But, what about the 0-1 km distance, where the highest number of distributed ruptures are observed (highest hazard)? In the discussion you deal with this point (lines 210-220), but it is not clear to me the message: are you suggesting the empirical-probabilistic approach from global data is not applicable in the 0-1 km distance? From your results, what is the suggestion to practitioners for the 0-1 km distance in a probabilistic approach? Please note that the cited references (Teran et al., 2015; Gold et al., 2015; Loukidis et al., 2009; Treiman, 2010) mostly refer to rupture zone widths that are much narrower than 1 km.

5) High values at 7-12 km distance and antithetic faults (lines 242-243). Interesting observation. Did you verify if the 7-12 km ruptures effectively correspond to antithetic faults? Please, can you cite the cases where they correspond?

6) Title. I suggest to modify into 'Distributed surface rupturing during normal faulting earthquakes ...'.

7) Data (shapefiles and Table 1): - 1915 Fucino M 7.1: the San Benedetto dei Marsi fault is considered distributed. I think you should reconsider this choice (maximum coseismic displacement was there). - 1954 Rainbow Mountain + 1954 Stillwater (Pezzopane and Dawson, 1996): The principal faults of the two events overlap for a large portion, but they have different traces (mainFaults in the shapefile). Is there something wrong? - 1980 Irpinia, 1997 Colfiorito, 2009 L'Aquila, 2016 Amatrice, 2016 Norcia: you should cite the source of the original rupture maps, as stated in the caption of Table 1. - 2009 L'Aquila: please note that there is a database on-line the line-work of which appears more detailed than that reported in your DB. See https://ingv.maps.arcgis.com/apps/webappviewer/index.html?id=05901efc172e489f8db4198bc43bf507

(it is already in Baize et al. 2019) - 1980 Gulf of Corinth: the association of the surface rupture to the second shock only (Feb. 25 M 6.4) or to both the first and second shocks (24 M 6.7 + 25) is not straightforward. See e.g., Hubert et al, 1996 EPSL. - 1980 Mammoth Lake: very complex event. How could you identify the main fault? - 1987 Edgecumbe: the rupture sections are not in the online DB. Did you use Baize et al. 2019?

8) Equation 1: specify that 'x' is in km; 'Log' should be 'Ln'.

9) Table 1, caption: cite Equation 1.

---

## Author Comment (AC1) · 15 Mar 2021

Conditional probability of distributed surface rupturing during normal faulting earthquakes Maria Francesca Ferrario1, Franz Livio1 1 Università dell'Insubria, Dipartimento di Scienza e Alta Tecnologia, Como, 22100, Italy

Response to reviewers: We wish to thank the two anonymous reviewers for their thoughtful comments, which helped in improve the quality of the manuscript. Here we provide a point-to-point answer to all the comments raised by reviewer 1.

Anonymous Referee #1

Here are the comments that I would like the authors to explain for readers with interest of PFDHA. 1. The conditional probability of the distributed rupture 1-1. Why did the authors update only the conditional probability, not distributed fault displacement attenuation relation?

Thanks for the comment. We integrated the Introduction section by adding the following paragraph: According to Youngs et al. (2003), the attenuation function for fault displacement, i.e., the term iii) of the general PFDHA function, can be split into two terms (Fig. 1):

Where k is the position of the site of interest, n is the seismogenic source, D is displacement at the site, d is a given displacement threshold, m is magnitude, r is distance from the principal fault to the site. The first term is the conditional probability that some amount of displacement occurs at site k, i.e., it represents the actual occurrence of distributed faulting (D > 0). The second term is the conditional probability of exceeding a given level of displacement (d). In this paper, we focus exclusively on the first term of Eq. 1; this choice was driven by the fact that surface faulting can be an exclusion criterion for some plants (e.g., nuclear power plants).

1-2. Petersen et al. (2011), who uses the power function, used linear interpolation as mentioned (l. 185-186). On the other hand, Youngs et al. (2003) and Takao et al. (2013), who use the exponential function used same to this paper, do not exclude the near range from the principal fault. Why do the authors need to exclude the range of 0-1km from the principal fault? 1-3. If your conditional probability excludes the distance range of 0-1km from the principal fault, I would like the authors to describe the calculation of the probability in this vicinity as Petersen et al. (2011).

In the first version of the manuscript, we excluded the 0-1 km from the principal fault purely for mathematical reasons. The ratio of conditional probability in the 0-0.5 km and 1-1.5 km is 7:1 and 14:1 for the hanging wall and footwall, respectively. As a consequence, including the near-field in the fitting gives birth to higher misfits in the

far-field. Following the comments by both reviewers, we now fit the data on the entire range in terms of distance. We tested different functional forms, including a piecewise regression (linear in the near field, exponential beyond 1 km), which is an approach similar to the one adopted by Petersen et al. (2011). Goodness of fit were substantially identical when trying different functional forms, while a higher impact is due to variations in the input data (i.e., we slightly modified the L'Aquila 2009 and Stillwater 1954 events, following the comments by Reviewer 2). We thus maintain the functional form originally selected, in agreement with the work of Youngs et al. (2003). We updated the methods section and all the figures in the manuscript and we provide the new fitting coefficients in Table 2.

2. As the authors mentioned (l. 217-220), conditional probability is obtained from the global data set. Is this the reason for the greater probability than that of Youngs et al. (2003)? In other words, is Youngs et al. (2003) used for the US PFDHA and is the conditional probability of this study used for the non-US PFDHA?

Following this input, we tested whether there is a systematic difference between US and non-US earthquakes; this also corresponds to the comparison of events analyzed by Youngs et al. (2003) and the additional events included in our paper. We found no clear difference both in terms of percentage of earthquakes showing distributed faulting vs no faulting for each distance class (Figure 2), nor for the conditional probability of faulting (Figure 3). As we mentioned in the paper, a different behavior between Japan and US ruptures has been identified in the literature (Inoue et al., 2018; Petersen and Chen, 2018; Suzuki and Annaka, 2018); the current research does not show such difference, but it may well be possible that this is due to the limited number of available earthquakes.

Here are the minor comments. 3. Title It is difficult to understand the detail contents from the title. 'normal earthquake' is expressions that is rarely seen for me. Does 'scaling relation' mean a conditional probability?

Following also the comments by Rev. 2, the title has been slightly changed to "Conditional probability of distributed surface rupturing during normal faulting earthquakes"

4. Typo? FDHA -> PFDHA? (l. 63)

Thanks, corrected.

5. Caption of Table 1 22 earthquakes may be 21 earthquakes.

Yes, we corrected our mistake.

5. Eq.1 (l. 112) Please add unit of x.

Done, x is in km.

6. Figure 6(a) Why is the yellow-colored range near the main fault in the figure different between the hanging wall and foot wall sides?

Thanks for spotting this, it was a graphic error. We now have fixed it.
* * *
$$P_{kn}(D > d|m,r) = P_{kn}(Slip|m,r) \times P_{kn}(D > d|m,r,Slip)$$

**Fig. 1.** equation 1

[Figure]

**Fig. 2.** percentage of earthquakes showing distributed faulting vs no faulting for each distance class, grouped according to geographical region (within and outside US).

[Figure]

**Fig. 3.** Conditional probability of faulting for the subset of event in the US and those outside US.

---

## Author Comment (AC2) · 15 Mar 2021

Response to reviewers We wish to thank the two anonymous reviewers for their thoughtful comments, which helped in improve the quality of the manuscript. Here we provide a point-to-point answer to all the comments raised by reviewer 2.

Anonymous Referee #2 Please, consider the following comments during the revision of the paper: 1) Nomenclature: Primary vs Principal. The reference literature on PFDHA uses this nomenclature: Principal and Distributed (Youngs et al 2003; Petersen et al., 2011). Though this is a very minor comment, it would be nice if all the specialized literature will use the same nomenclature.

Thanks for the comment. We totally agree and now use "principal" instead of primary.

2) Method (gridding) (Lines 100-102, Fig 2a). In order to make the results reproducible, can you be more detailed in describing the geometry of the grid and the method for calculating distances? Did you consider the same maximum distance from PF for all the events? How far from PF (20 km?)? Which criterion guided the choice?

We considered a distance range of 20 km in the hanging wall and 15 km in the footwall for all the events; this choice was guided by the maximum distance of observed DF in our dataset, i.e., in the bins 15-15,5 km in the hanging wall and 12-12,5 km in the footwall, respectively. We realize that the figures in the paper had different x-axis limits, so now we made them uniform and show the ranges of 0-20 km in the hanging wall and 0-15 km in the footwall.

Was the grid (and the sides of the squares) always horizontal/vertical? Or rotate with fault strike? PF-distance: is that the shortest distance between the PF line and the centre of the cell? Adding a real case in Fig. 2a can be explicative and help the reader.

The grid was computed using the "Euclidean distance" tool in ArcMap. The tool internally transforms the source layer into a raster and Euclidean distance is calculated from the center of the source cell to the center of each of the surrounding cells. The grid is built along the x, y coordinates, so it is always horizontal/vertical. Concerning DF, the geometry of the ruptures is originally a linear shapefile; we converted them into a grid, again using the 500 m size and compute the distance from the center of the cell to the center of the closest cell along the principal fault. We better addressed the methodological implications in the discussion section (see below response to point 4). Since the 500 x 500 m grid size is quite coarse, we believe that this source of uncertainty is much lower than other causes, in particular the resolution and quality of original maps or input data.

3) Extrapolation of regressions (lines 174-175). Why did you extrapolate away from observations?

We modified the sentence because it was misleading, in particular "extrapolation" was a bad word choice. The text is changed as follows: Another aspect worth mentioning is that real data are constrained down to probabilities as low as ca. 10-2 (see "No data" field in Fig. 3a); this lower threshold is constrained by the number of pixels for each distance bin, which in turn depends on the grid size of the analysis and the length of the principal fault. Probabilities lower than this threshold derive from the averaging of the earthquakes actually producing DF with those not producing DF.

4) Role of dataset age and M. You found no systematic bias between the pre- and post-2000 datasets (lines 179-181). Looking at your data, I agree with this observation. But I suspect that you do not see the differences in the analysis because of the small number of modern data compared to older data and the coarse grid size, which smooths differences. I suspect that this can influence the possible dependency on M, as well. I think this point (possible bias due to methodology of analysis) should be addressed in the discussion.

We added a few sentences in the discussion section to address the issue of grid size resolution. We also highlight that the acquisition of future datasets is critical for assessing eventual systematic biases and will allow to test different grid sizes.

4) Distributed faulting in the near-field (< 1 km). The regressions are cut at distances shorter than 1 km for mathematical reasons that I can understand. But, what about the 0-1 km distance, where the highest number of distributed ruptures are observed (highest hazard)? In the discussion you deal with this point (lines 210-220), but it is not clear to me the message: are you suggesting the empirical-probabilistic approach from global data is not applicable in the 0-1 km distance? From your results, what is the suggestion to practitioners for the 0-1 km distance in a probabilistic approach? Please note that the cited references (Teran et al., 2015; Gold et al., 2015; Loukidis et

al., 2009; Treiman, 2010) mostly refer to rupture zone widths that are much narrower than 1 km.

Following the comments by both reviewers, we now fit the data on the entire range in terms of distance. We tested different functional forms, including a piecewise regression (linear in the near field, exponential beyond 1 km), which is an approach similar to the one adopted by Petersen et al. (2011). Goodness of fit were substantially identical when trying different functional forms, while a higher impact is due to variations in the input data (i.e., we slightly modified the L'Aquila 2009 and Stillwater 1954 events, following the comments by Reviewer 2). We thus maintain the functional form originally selected, in agreement with the work of Youngs et al. (2003). We updated the methods section and all the figures in the manuscript and we provide the new fitting coefficients in Table 2.

5) High values at 7-12 km distance and antithetic faults (lines 242-243). Interesting observation. Did you verify if the 7-12 km ruptures effectively correspond to antithetic faults? Please, can you cite the cases where they correspond?

We checked the events and accordingly added the following sentence: "As a first approximation, this pattern is quite evident for the 1980 Irpinia and 2016 Norcia earthquakes, and more subdued for the 1954 Fairview Peak, 1959 Hebgen Lake, 1970 Geidz and 1987 Edgecumbe events". Nevertheless, more efforts are needed to confirm this hypothesis, because in some instances synthetic splays are present as well.

6) Title. I suggest to modify into 'Distributed surface rupturing during normal faulting earthquakes'.

Following also the comments by Rev. 1, the title has been slightly changed to "Conditional probability of distributed surface rupturing during normal faulting earthquakes"

7) Data (shapefiles and Table 1):

Thanks for the very detailed comments on several of the events. Following the suggestions, we modified the trace/interpretation for some of the considered earthquakes; below more complete answers to each comment. In some other cases (e.g., Fucino 1915; Gulf of Corinth 1981; Mammoth Lake 1980) we maintain the earthquakes in our dataset despite a possibly higher epistemic uncertainty because, as discussed in the manuscript, there is a clear lack of case histories. We address this issue in the discussion section.

- 1915 Fucino M 7.1: the San Benedetto dei Marsi fault is considered distributed. I think you should reconsider this choice (maximum coseismic displacement was there).

We acknowledge that the interpretation of the Fucino case history is not straightforward and the epistemic uncertainty is higher than for other events. We decided to don't change our interpretation for the following reasons. We based our interpretation on published literature and the morphotectonic setting. Contemporary eyewitnesses attest to surface faulting occurring on several fault strands after the 1915 earthquake, but the original sources are not adequate to categorize them in primary and distributed. From the morphotectonic point of view, all the considered faults have a similar surficial expression, characterized by scarps bounding an intramountain basin. We follow the work by Galadini & Galli (Tectonophysics 308, 143–170, 1999), which claim that the max displacement in the 1915 earthquake was on the Marsicana Highway Fault (see Figures 3 and 4). The high degree of uncertainty for this event is also attested by the fact that surface faulting along the Magnola fault is mainly derived from interpretation of paleoseismological data (Galli et al., 2012, Bollettino di Geofisica Teorica ed Applicata 53, 4, 435-458).

- 1954 Rainbow Mountain + 1954 Stillwater (Pezzopane and Dawson, 1996): The principal faults of the two events overlap for a large portion, but they have different traces (mainFaults in the shapefile). Is there something wrong?

The Rainbow Mountain and Stillwater earthquakes occurred on 06/07/1954 and 23/08/1954, respectively. They ruptured the same fault segment, and ca. 12 km of

the surface rupture of the two events are overlapping. In this sector, ruptures are discontinuous. We changed the trace of the Stillwater principal fault, making it coincident with the Rainbow Mt. one in the overlapping sector.

- 1980 Irpinia, 1997 Colfiorito, 2009 L'Aquila, 2016 Amatrice, 2016 Norcia: you should cite the source of the original rupture maps, as stated in the caption of Table 1.

Correct, we changed Table 1 by adding the original references instead of Ferrario & Livio, 2018.

- 2009 L'Aquila: please note that there is a database on-line the linework of which appears more detailed than that reported in your DB. See https://ingv.maps.arcgis.com/apps/webappviewer/index.html?id=05901efc172e489f8db4198bc43bf507 (it is already in Baize et al. 2019)

Ok. We now use the dataset included in Baize et al. (2019).

- 1980 Gulf of Corinth: the association of the surface rupture to the second shock only (Feb. 25 M 6.4) or to both the first and second shocks (24 M 6.7 + 25) is not straightforward. See e.g., Hubert et al, 1996 EPSL.

For the interpretation of the Gulf of Corinth events we followed the work by Jackson et al. (1982), which analyzed teleseismic, local seismic data, surface faulting effects and the geomorphological setting. The authors conclude "that most of the faulting on land was due to the second shock [i.e., Feb. 25] and that the first earthquake [i.e., Feb 24] occurred on an offshore fault". We acknowledge that the association of ground breaks with their causative event is a tricky task and we highlight this fact in the discussion section. We believe that the occurrence of several events in a short time interval brings a significant degree of epistemic uncertainty. For case histories like the 1981 Gulf of Corinth sequence it may be difficult, if not impossible at all, identify the ground ruptures for each event; for older or less documented cases, it may be difficult even identify if single or multiple earthquakes occurred. PFDHA is a young science

relying on empirical data, thus only enlarging the dataset of case histories will allow a reduction in the epistemic uncertainty. Since in our paper we deal with the conditional probability of faulting as a function of distance, and we do not address the influence of magnitude, we believe that by including the February 25, 1981 earthquake as mapped by Jackson et al. (1982) does not significantly alter the output of the results. If the ground breaks were erroneously attributed to the February 25 earthquake, the case history should be seen as the cumulative effect of the February 24 and 25 events. Current data does not allow to validate this hypothesis.

- 1980 Mammoth Lake: very complex event. How could you identify the main fault?

Surface faulting following the Mammoth Lake event was ca. 16 km long, offsets were generally centimetric (max 20 cm) and the pattern in map view is quite complex. We defined the principal fault based on the strike of ground breaks and their continuity, as mapped by Pezzopane & Dawson (1996). We agree that the definition of the principal fault for this event is quite tricky; on the other hand, we want to retain the highest possible number of case histories.

- 1987 Edgecumbe: the rupture sections are not in the online DB. Did you use Baize et al. 2019?

Correct, we did not include the Edgecumbe ruptures because already published.

8) Equation 1: specify that 'x' is in km; 'Log' should be 'Ln'.

Done.

9) Table 1, caption: cite Equation 1.

Done. It is now Eq. 2.
* * *
[Figure]

**Fig. 1.** map of the surface faulting following the 1915 Fucino earthquake. From Amoroso et al. (2016).

**Fig. 2.** 1915 vertical offset along the different fault branches (MHF: Marsicana Highway Fault, SBGF: San Benedetto – Gioia dei Marsi Fault), from Galadini & Galli (1999).